# DEFORMABLE CAPSULES FOR OBJECT DETECTION

## ABSTRACT

Capsule networks promise significant benefits over convolutional networks by storing stronger internal representations, and routing information based on the agreement between intermediate representations' projections. Despite this, their success has been mostly limited to small-scale classification datasets due to their computationally expensive nature. Recent studies have partially overcome this burden by locally-constraining the dynamic routing of features with convolutional capsules. Though memory efficient, convolutional capsules impose geometric constraints which fundamentally limit the ability of capsules to model the pose/deformation of objects. Further, they do not address the bigger memory concern of class-capsules scaling-up to bigger tasks such as detection or large-scale classification. In this study, we introduce deformable capsules (*DeformCaps*), a new capsule structure (*SplitCaps*), and a novel dynamic routing algorithm (*SE-Routing*) to balance computational efficiency with the need for modeling a large number of objects and classes. We demonstrate that the proposed methods allow capsules to efficiently scale-up to large-scale computer vision tasks for the first time, and create the first-ever capsule network for object detection in the literature. Our proposed architecture is a one-stage detection framework and obtains results on MS COCO which are on-par with state-of-the-art one-stage CNN-based methods, while producing fewer false positive detections.

## 1 INTRODUCTION

Capsule networks promise many potential benefits over convolutional neural networks (CNNs). These include practical benefits, such as requiring less data for training or better handling unbalanced class distributions (Jiménez-Sánchez et al., 2018), and important theoretical benefits, such as building-in stronger internal representations of objects (Punjabi et al., 2020), and modeling the agreement between those intermediate representations which combine to form final object representations (e.g. part-whole relationships) (Kosiorek et al., 2019; Sabour et al., 2017). Although these benefits might not be seen in the performance metrics (e.g. average precision) on standard benchmark computer vision datasets, they are important for real-world applications. As an example, it was found by Alcorn et al. (2019) that *CNNs fail to recognize 97% of their pose space*, while capsule networks have been shown to be far more robust to pose variations of objects (Hinton et al., 2018); further, real-world datasets are not often as extensive and cleanly distributed as ImageNet or MS COCO.

These benefits are achieved in capsule networks by storing richer vector (or matrix) representations of features, rather than the simple scalars of CNNs, and dynamically choosing how to route that information through the network. The instantiation parameters for a feature are stored in these capsule vectors and contain information (e.g. pose, deformation, hue, texture) useful for constructing the object being modeled. Early studies have shown strong evidence that these vectors do in fact capture important local and global variations across objects' feature components (or parts) within a class (Punjabi et al., 2020; Sabour et al., 2017). Inside their networks, capsules dynamically route their information, seeking to maximize the agreement between these vector feature representations and the higher-level feature vectors they are attempting to form.

Despite their potential benefits, many have remained unconvinced about the general applicability of capsule networks to large-scale computer vision tasks. To date, no capsule-based study has achieved classification performance comparable to a CNN on datasets such as ImageNet, instead relegated to smaller datasets such as MNIST or CIFAR. Worse still, to the best of our knowledge, *no capsule network has shown successful results in object detection*, a very important problem in computer

vision, robotics, and medical imaging. Now, the argument can be made that standard benchmark datasets such as ImageNet or MS COCO likely contain majority of objects in that 3% range of usual poses, and thus CNNs will appear to perform extremely well when measured in terms of accuracy, stripping capsule networks of one of their largest advantages. However, until capsule networks can perform on-par with CNNs on these typical object poses, few will care about the benefits of stronger internal representations and better generalization to unseen poses.

**Summary of Our Contributions:** (1) We propose the first ever capsule-based object detection framework in the literature. Our network is a one-stage (single-shot) architecture, where objects are both localized and classified using capsules, and can perform on-par with the state-of-the-art CNNs on a large-scale dataset (MS COCO). (2) We address the geometric constraint of convolutional capsules (and locally-constrained routing) by introducing *deformable capsules*, where parent capsules learn to adaptively sample child capsules, effectively eliminating rigid spatial restrictions while remaining memory efficient. (3) We design a new capsule-based prediction head structure, *SplitCaps*, which reformulates the projections of an objects' instantiation parameters, presence, and class, eliminating the previous dimensional increase of capsules by the number of classes. This crucial addition enables the training of capsule networks on large-scale computer vision datasets for the first time in the literature. (4) To route information across *SplitCaps'* unique structure, we introduce a novel Squeeze-and-Excitation inspired dynamic routing algorithm, *SE-Routing*, which seeks to maximize agreement between child capsule projections, without the need of iterative loops

## 2 DEFORMABLE CAPSULES: FIXING LOCALLY-CONSTRAINED DYNAMIC ROUTING

The capsule network architecture proposed by Sabour et al. (2017) acted on global information, where digit capsules represented the pose and presence of digits in an image regardless of spatial location. The information from all children in the previous layer was sent to every parent in the following layer, weighted via the routing coefficients found in a cosine similarity routing algorithm. While this proved to be a highly-effective strategy, it was also computationally expensive, limiting its use to *only* small-scale datasets. Recent works attempted to scale up capsule networks to larger problems such as biomedical image segmentation (LaLonde & Bagci, 2018) or action detection in video (Duarte et al., 2018) by using convolutional capsules and locally-constraining the routing algorithm. Although efficient solutions were presented in those studies, the representation power of capsule networks was fundamentally limited due to imposing local constraints. This is because convolutions, by design, have a fixed geometric structure (Dai et al., 2017), and such a geometric constraint significantly inhibits capsules' ability to model part-whole relationships, relying on parts of objects to fall within a fixed local grid. Therefore, it is unreasonable to expect a capsule to effectively represent the pose and deformations of an object when the information related to the parts of that object are locked into a fixed spatial relationship.

In this study, we propose to effectively solve this aforementioned problem by introducing a method that balances efficiency with the ability for a capsule to represent any pose and deformation of an object (i.e. where child capsules can be found in different spatial relationships to one another for the same parent). In such a formulation, global information is not explicitly required, but it does require parent capsules to have more flexibility over which child capsules they draw information from. Our proposed solution is *deformable capsules*. The idea behind the proposed algorithm is simple: if parent capsules are supposed to capture common deformations of the objects they represent within their vectors, then the choice of which children to aggregate information from must be handled in a deformable manner as well. Deformable capsules allow parents to adaptively gather projections from a non-spatially-fixed set of children, and thus *effectively and efficiently* model objects' poses.

To achieve this overall goal, we follow the same efficient convolutional capsule paradigm, where projection vectors are formed via a convolution operation with a kernel centered on the parent capsules' spatial location, but now we learn an additional set of weights for each parent capsule. These learnable weights are the same shape as each parent's kernel, and represent the offset values for the spatial sampling of child capsules for that parent. Based on the child capsule representation vectors in the previous layer, these weights learn which children a parent capsule should adaptively sample from for a given input image. Dynamic routing then determines how to weight the information coming from each of these children based on their agreement for each projected parent.

Let us give a concrete example to better illustrate the parameter savings of this technique. Given a $H = 128$ by $W = 128$ grid of child capsules with $c_i = 32$ capsule types of $a_i = 8$ atoms each, being routed to a set of $c_j = 10$ parent capsule types of $a_j = 16$ atoms each, the fully-connected capsules of Sabour et al. (2017) would require $H \times W \times c_i \times a_i \times c_j \times a_j \Rightarrow 128 \times 128 \times 32 \times 8 \times 10 \times 16 \approx 671M$ parameters for this layer alone (assuming the goal is classification, with detection requiring a multiplicative increase by the detection grid size). Instead, using our proposed deformable capsules with a $k^2 = 5^2$ kernel, we only require $2 \times k \times k \times a_i \times c_j \times a_j \Rightarrow 2 \times 5 \times 5 \times 8 \times 10 \times 16 \approx 64K$ parameters. Convolutional capsules with locally-constrained routing require 32K parameters, not needing the additional spatial offsets kernel, but as mentioned above, they are fundamentally limited in the poses and deformations that they can represent. In our experiments, we found deformable capsules to converge faster and to much higher performance than convolutional capsules.

## 3 OBJECTS AS CAPSULES: *SplitCaps* WITH *SE-Routing*

We propose a novel one-stage (single-shot) capsule network architecture for object detection, called *DeformCaps*, where objects are detected, classified, and modeled with capsules. Our overall network architecture, shown in Fig. 1, is built upon *CenterNet* by Zhou et al. (2019a) who proposed to represent objects in images as scalar point values located at the center of their bounding boxes. The authors then regress the remaining characteristics of the object (e.g. height, width, depth) for each center-point detected. In our work, we follow the same center-point detection paradigm, but represent our objects with capsule vectors instead. Since several recent studies have found utilizing a CNN backbone before forming capsule types to be beneficial to overall performance (Duarte et al., 2018; Kosiorek et al., 2019; Tsai et al., 2020), we adopt the preferred backbone of CenterNet, DLA-34 (Yu et al., 2018), as this gave the best trade-off between speed and accuracy. Features extracted from the backbone are sent to our capsule object detection head and a bounding box regression head. In order to perform object detection using capsules, we introduce a new capsule structure, called *SplitCaps*, composed of class-agnostic and class-presence capsules, and a novel routing algorithm, called *SE-Routing*.

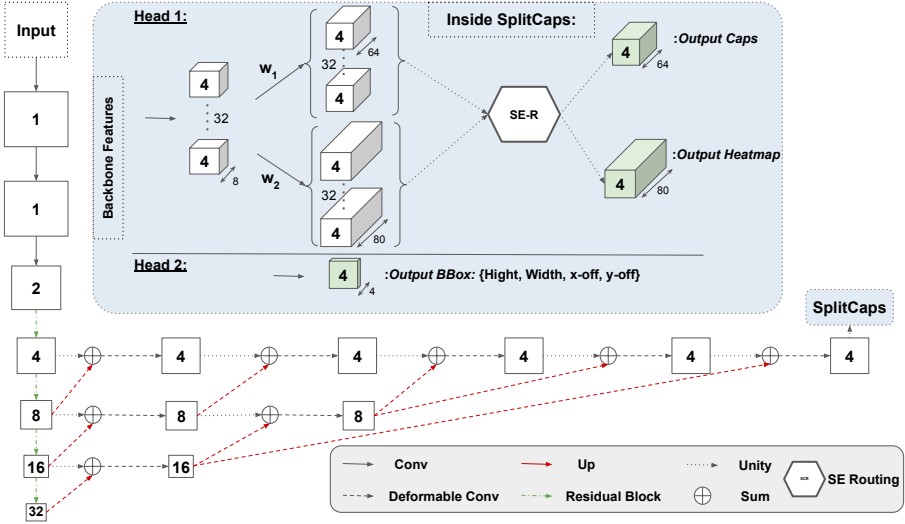

Figure 1: Deformable capsule architecture for object detection.

### 3.1 SPLITCAPS: CLASS-AGNOSTIC CAPSULES AND CLASS-PRESENCE CAPSULES

As discussed in Section 2, the original CapsNet by Sabour et al. (2017) was extremely expensive in computation and our proposed deformable capsules is a best possible solution to balance non-rigid deformations of objects while remaining memory efficient. However, *there is a more significant memory hurdle capsule networks must overcome when scaling up to large-scale datasets*, such as MS COCO. In their current implementation, capsule networks represent each class with its own parent

capsule vector. On small scale classification datasets (and using deformable routing), this is not an issue; it amounts to $2 \times k \times k \times a_i \times c_j \times a_j$ parameters and $N \times c_i \times c_j \times a_j \times 4$ bytes to store the intermediate representations to be routed for the parents, where $c_j$ is usually around 10 classes to represent. Let us suppose we have $5 \times 5$ kernels with 32 input capsule types of 8 atoms per capsule, 10 output capsule types of 16 atoms per capsule, and a batch size of 32. In total, we would have $2 \times 5 \times 5 \times 8 \times 10 \times 16 = 64K$ parameters and $32 \times 32 \times 10 \times 16 \times 4 \approx 655$ KB.

When we scale this up to object detection, we now need to store representations *for every possible object location*, and for MS COCO we need to *represent 80 possible classes*. This gives us $2 \times k \times k \times a_i \times c_j \times a_j \Rightarrow 2 \times 5 \times 5 \times 8 \times 80 \times 16 = 512K$ parameters and $N \times H \times W \times c_i \times c_j \times a_j \times 4$ bytes $\Rightarrow 32 \times 128 \times 128 \times 32 \times 80 \times 16 \times 4 \approx 86$ GB for the intermediate representations, where we assume the output grid of detections is $128 \times 128$ with a single detection (i.e. bounding box) predicted per class per location. The problem is not any better for large-scale classification datasets such as ImageNet either, where we lose the grid of predictions but grow to 1000 classes, which would require $2 \times 5 \times 5 \times 8 \times 1000 \times 16 = 6.4M$ parameters and $32 \times 32 \times 1000 \times 16 \times 4 \approx 66$ GB for the intermediate representations. Clearly, with most GPU memories limited to 12–24 GB, a solution is needed to be found for capsule networks to scale up to larger-scale computer vision tasks.

To overcome this issue, we propose a new type of capsule architecture, *SplitCaps*, to more efficiently scale capsule networks to large-scale computer vision tasks. SplitCaps contains two parent capsule types, each with a different number of atoms per capsule, for each location of the detection grid. As before, the idea is to balance efficiency with the ability to learn powerful representations. Towards this goal, SplitCaps proposes to divide up between its two parent capsules the tasks of *(i)* learning the instantiation parameters necessary to model the possible variations of an object and *(ii)* predicting which classes of objects are present in a given input. The first capsule type we refer is our class-agnostic object instantiation capsules, and the second we refer is class presence capsules.

**Class-agnostic object instantiation capsules:** The purpose of these capsules is similar to those in previous works: model the possible variations (in pose, deformation, texture, etc.) of objects within a vector of instantiation parameters, the span of which should cover all possible variations for that object at test (hence why capsules are better than CNNs at generalizing to unseen poses). While previous capsule networks did this in a class-wise manner, we argue such a formulation is not required and possibly it is redundant. Many variations (e.g. rotation, skew, stroke thickness) may be class-independent, and thus to model these variations class-wise would require repetition across each capsule type. Instead, we propose to model all classes within a single capsule type (i.e. class-agnostic). In this way, while class-dependent variations would each require their own dimensions of the vector, class-independent variations can each be modeled along a single dimension for all possible objects. Since it is reasonable to assume there will be at least some class-specific variations, we increase the default capsule vector dimension from 16 to 64 to accommodate for possible class-specific instantiation parameters.

During training, if an object is present at a given spatial location, the 64-dimensional capsule vector for that location is fed to a reconstruction regularization sub-network to construct the mask of that object as similar to the reconstruction regularization used by Sabour et al. (2017) for classification. This sub-network is a relatively small and fast addition: a set of three ReLU-activated $1 \times 1$ convolutional layers with 256 filters each, followed by a final sigmoid-activated $1 \times 1$ convolution with $N = n^2 = 28^2 = 784$ filters, before reshaping outputs to $n \times n$. Since objects' scales vary dramatically, we scale normalize all objects' ground-truth masks to be $28 \times 28$ (by following He et al. (2017)). Supervised training is conducted by computing the Dice loss (Milletari et al., 2016) between the predicted reconstruction, $r$, and the object's mask, $m$

$$\mathcal{L}_r = \frac{2 \sum_i^N r_i m_i}{\sum_i^N r_i^2 + \sum_i^N m_i^2}, \tag{1}$$

where $\mathcal{L}_r$ is used to provide a regularization signal to the instantiation parameters being learned.

**Class presence capsules:** The class presence capsules attempt to model which classes of objects are present in the input at each spatial location, if any. We accomplish this by setting the atoms per capsule to the number of classes being represented (i.e. 80 for MS COCO). Just as Hinton et al. (2018) separately modeled pose (with a matrix) and activation (with a scalar), this 80-dimensional vector can be viewed as a class-dependent set of activation values. The activation values are then passed through a sigmoid function and thresholded; if one or more activation values are above the

threshold, an object is determined to be at that spatial location with the strongest activated dimension determining the class label.

In order to produce a smooth loss function during training, we create a ground-truth heatmap by fitting a Gaussian distribution rather than a single point, to the center-point of each object's bounding box, with variance proportional to the size of the box following Zhou et al. (2019a). More specifically, we create a heatmap $H \in [0,1]^{\frac{X}{d} \times \frac{Y}{d} \times K}$ containing each down-scaled ground truth center-point $\tilde{p} = \left(\frac{p_x}{d}, \frac{p_y}{d}\right)$ for class $k \in K$ using a Gaussian kernel $H_{xyk} = \exp\left(-\frac{(x-\tilde{p}_x)^2 + (y-\tilde{p}_y)^2}{2\sigma_p^2}\right)$, where $d$ is the amount of downsampling in the network and $\sigma_p$ is an object-size-adaptive standard deviation (Law & Deng, 2018). In the case of overlapping Gaussians, we take the element-wise maximum. To handle the large class imbalance between objects and background in our heatmaps, we use a penalty-reduced pixel-wise logistic regression with a focal loss (Lin et al., 2017):

$$\mathcal{L}_h = \frac{-1}{P} \sum_{xyk} \begin{cases} (1 - \hat{H}_{xyk})^\alpha \log(\hat{H}_{xyk}) & \text{if } H_{xyk} = 1, \\ (1 - H_{xyk})^\beta (\hat{H}_{xyk})^\alpha \log(1 - \hat{H}_{xyk}), & \text{otherwise;} \end{cases} \tag{2}$$

where $\alpha, \beta$ are hyper-parameters of the focal loss, $P$ is the number of center-points in the input, used to normalize all positive focal loss instances to 1 (Zhou et al., 2019a). We use $\alpha = 2$ and $\beta = 4$ in all our experiments, following Law & Deng (2018). At test, to efficiently retrieve the object's exact center, we run a $3 \times 3$ max-pooling over the thresholded spatial map.

To predict the height and width of the bounding boxes of objects and recover the $x, y$ offsets needed to map back to the upscaled image dimensions, we follow the same formulation as Zhou et al. (2019a) and pass the backbone features through a $3 \times 3$ convolutional layer with 256 feature maps, then a $1 \times 1$ convolutional layer with 2 feature maps. These layers predict the local offset, $\hat{O} \in \mathbb{R}^{\frac{W}{d} \times \frac{H}{d} \times 2}$, and size prediction, $\hat{S} \in \mathbb{R}^{\frac{W}{d} \times \frac{H}{d} \times 2}$, for each center-point and are supervised by

$$\mathcal{L}_o = \frac{1}{P} \sum_p \left| \hat{O}_{\tilde{p}} - \left(\frac{p}{d} - \tilde{p}\right) \right| \quad \text{and} \quad \mathcal{L}_s = \frac{1}{P} \sum_p \left| \hat{S}_p - (x_2 - x_1, y_2 - y_1) \right|, \tag{3}$$

respectively. Our final objective function is thus defined as $\mathcal{L} = \mathcal{L}_h + \lambda_r \mathcal{L}_r + \lambda_s \mathcal{L}_s + \lambda_o \mathcal{L}_o$. We keep $\lambda_s = 0.1$ and $\lambda_o = 1$ as done in Zhou et al. (2019a), and set $\lambda_r = 0.1$ initially, then step up to $\lambda_r = 2.0$ at the half-way in training.

## 3.2 SE-ROUTING: SPLITCAPS NEW ROUTING ALGORITHM

As a core component of capsule networks, dynamic routing seeks to maximize the agreement between child capsule projections for parent capsules and to fully-leverage the richer representations being stored. Since SplitCaps introduces a unique capsule head structure, where instantiation parameters and activations are split across different capsule types, previous dynamic routing algorithms can no longer be directly applied. To overcome this, we propose a new dynamic routing algorithm that takes inspiration from Squeeze-and-Excitation networks (Hu et al., 2018), which we call *SE-Routing*, and illustrated in Fig. 2.

Previously proposed dynamic routing algorithms (e.g. Sabour et al. (2017) and Hinton et al. (2018)) were typically iterative, requiring a hand-tuned loop of routing iterations, which proved to be slow and temperamental in practice. Different studies found different numbers of iterations to be effective, and one meta-study of five different iterative dynamic routing algorithms found them all to be largely ineffective (Paik et al., 2019). To avoid this pitfall, we propose a new routing algorithm to dynamically assign weights to child capsule projections based on their agreement, computed in a single forward pass using a simple gating mechanism with sigmoid activation. Unlike Sabour et al. (2017) which uses a routing softmax to force a one-hot mapping of information from each child to parents, our proposed SE-Routing learns a non-mutually-exclusive relationship between children and parents to allow multiple children to be emphasised for each parent.

**Creating child capsule projection descriptors (squeeze):** Following the Squeeze-and-Excitation paradigm, we first must compute the squeeze (i.e. a set of descriptors which summarize relevant information about each feature) to create a set of child capsule projection descriptors. In Hu et al. (2018), the authors proposed to use the global average activation of each channel with the goal of modeling channel interdependencies. In this study, our goal is to maximize the agreement

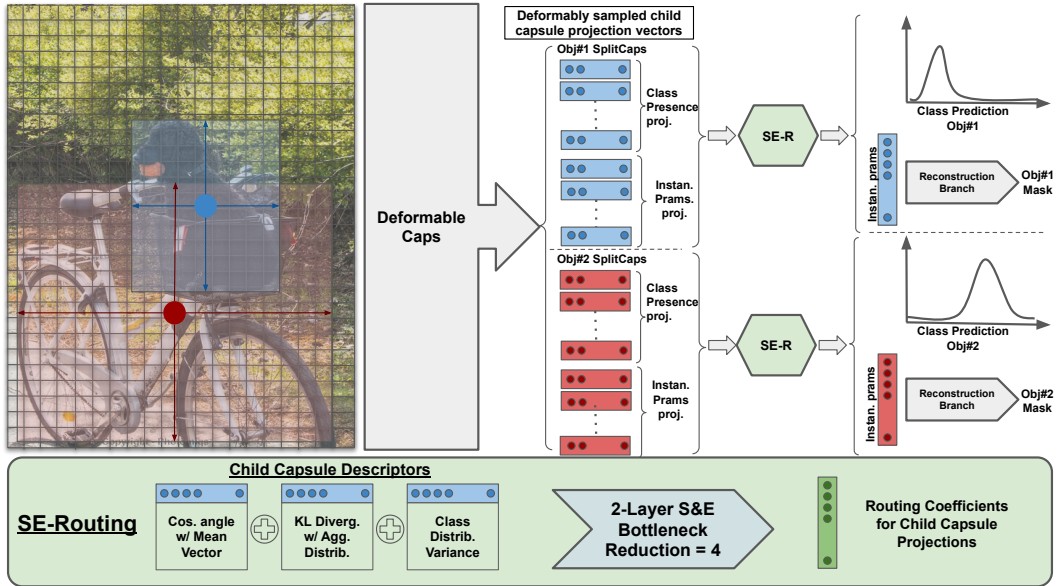

Figure 2: Proposed deformable SplitCaps formulation which, in one-shot, localizes all objects to their centers, determines their classes, and models their instantiation parameters in two parallel parent capsule types. Information is dynamically routed from children to parents by passing three chosen child capsule descriptors through a two-layer Squeeze-and-Excitation (S&E) bottleneck.

between child projections, for both the instantiation parameters and class presence of the object being modeled. With that motivation, we compute three separate descriptors which are fed into the excitation phase of the routing: *(i)* cosine angle between the mean projection vector and each child's projection, which captures *object instantiation agreement*; *(ii)* Kullback–Leibler (KL) divergence of each child's predicted class distribution and an aggregated distribution, which captures *class presence agreement*; and *(iii)* variance of each child's predicted class distribution, which captures *class presence uncertainty*.

The cosine angle descriptor, $\boldsymbol{a}$, is calculated in a similar manner to Sabour et al. (2017). A mean projection vector, $\tilde{\boldsymbol{u}} = 1/N \sum_i^N \hat{\boldsymbol{u}}_i$, is first computed using the set of child capsule projections, $\hat{\boldsymbol{U}} = \{\hat{\boldsymbol{u}}_1, \hat{\boldsymbol{u}}_2, ..., \hat{\boldsymbol{u}}_N\}$. Then we compute a set of cosine angles between each individual projection and this mean, $\boldsymbol{a} = \{a_1, a_2, ..., a_N\}$, where $a_i = (\tilde{\boldsymbol{u}} \cdot \hat{\boldsymbol{u}}_i)/(|\tilde{\boldsymbol{u}}| \cdot |\hat{\boldsymbol{u}}_i|)$.

In a similar fashion, we compute a KL divergence descriptor, $\boldsymbol{b}$, by first creating an aggregate object class distribution. To create this aggregate distribution, we follow the work of Clemen & Winkler (1999), insofar as each child capsule type is treated as an expert giving its prediction about the true underlying class distribution. First, we compute a simple linear opinion pool (Stone, 1961), $p(\tilde{z}) = \sum_i^N \sigma_s(\boldsymbol{z}_i)/N$, where $p(\tilde{z})$ is the aggregated probability distribution, $\boldsymbol{Z} = \{\boldsymbol{z}_1, \boldsymbol{z}_2, ..., \boldsymbol{z}_N\}$ is the set of child class presence projection vectors, and $\sigma_s(\boldsymbol{z}_i)_j = e^{\boldsymbol{z}_{ij}} / \sum_k^K e^{\boldsymbol{z}_{ik}}$ for $j = \{1, ..., K\}, i = \{1, ..., N\}$ is the softmax function used to transform projection vectors into normalized probability distributions over the $K$ classes. Then, we measure the agreement between each child's predicted distributions, $\sigma_s(\boldsymbol{z}_i)$, and the aggregate distribution, $p(\tilde{z})$, as the KL divergence between them $b_i = \sum_k^K p(\tilde{z}_k) \log(p(\tilde{z}_k)/\sigma_s(\boldsymbol{z}_i)_k)$.

Lastly, we take our child capsules' predicted distributions, $\sigma_s(\boldsymbol{z}_i)$, and compute their variance to estimate the uncertainty each child has: $c_i = \sum_k^K (\sigma_s(z_i)_k - \sum_k^K \sigma_s(z_i)_k)^2$. Our three sets of descriptors are efficiently computed for all capsules simultaneously (i.e. for entire batch and across spatial locations) on GPU in parallel with matrix operations. They are then concatenated, $\boldsymbol{s} = \boldsymbol{a} \oplus \boldsymbol{b} \oplus \boldsymbol{c}$, and fed to the excitation layers of our routing mechanism (Fig. 2).

**Determining routing coefficients (excitation):** The excitation stage of the SE-Routing algorithm has the task of learning a mapping from the concatenated set of capsule descriptors, $\boldsymbol{s}$, into a set

of routing coefficients for the child capsule projections. Since parents capsules types are no longer different classes in our formulation, but rather two separated aspects of modeling objects, we compute a single set of routing coefficients at each spatial location for both parents. Formally, this mapping is computed as $r = \sigma(W_2\delta(W_1s))$, where $W_1 \in \mathbb{R}^{\frac{3N}{t} \times 3N}$, $W_2 \in \mathbb{R}^{N \times \frac{3N}{t}}$, $\delta$ is the ReLU activation function, $\sigma$ is the sigmoid activation function, and $t$ is the reduction ratio used to form this mapping into a two fully-connected (FC) layer bottleneck. A brief note: although excitation routing has interesting parallels to self-attention (e.g. dynamically conditioned on the input), our learned mapping is non-mutually-exclusive, while self-attention and CapsNet's dynamic routing both rely on applying a softmax function over outputs.

Finally, with the determined routing coefficients $r = \{r_1, r_2, ..., r_N\}$, we can compute the output of the SplitCaps detection head. Projection vectors from each child to each parent are computed using the proposed deformable capsules (as described in Section 2). These projections are then combined and weighted by the routing coefficients to form the final parent capsules. These final parents contain the instantiation parameters, $v_{obj} = \sum_i^N r_i\hat{u}_{obj|i}$, and class presence, $v_{cls} = \sum_i^N r_i\hat{u}_{cls|i}$, of any objects being represented at the given spatial location within the detection grid.

## 4 DEFORMABLE CAPSULES ON MS COCO

We evaluated our deformable capsule object detection framework on the MS COCO dataset (Lin et al., 2014), which contains 118K training, 5K validation and 20K hold-out testing images. Average precision (AP) is reported over all IOU thresholds and at thresholds 0.5 (AP50) and 0.75 (AP75). We followed the training procedure proposed in Zhou et al. (2019a), training on $512 \times 512$ pixel inputs, yielding $128 \times 128$ detection grids, using random flip, random scaling (between 0.6 to 1.3), cropping, and color jittering as data augmentation, and Adam (Kingma & Ba, 2014) to optimize our objective function. Due to limited compute resources, we initialized the backbone network weights from CenterNet and only train for 40 epochs with a batch size of 12 and learning rate of 5e-4 with $5\times$ drops at 5, 15, and 25 epochs. Longer training would likely yield superior results, as found by Zhou et al. (2019a) who obtained better results for CenterNet when increasing from 140 to 230 epochs.

In Table 1, we provide results of our proposed deformable capsule network with and without flip and multi-scale augmentations following Zhou et al. (2019a). Inference time on our hardware (Intel Xeon E5-2687 CPU, Titan V GPU, Pytorch 1.2.0, CUDA 10.0, and CUDNN 7.6.5) was consistent with those reported by Zhou et al. (2019a).[1] While DeformCaps performs slightly worse than CenterNet in terms of AP, it does so while producing far fewer false positive detections, as shown in Table 2 in Appendix A.2. For ablations, we trained a version of DeformCaps which replaces the proposed deformable capsules with the standard locally-constrained convolutional capsules (*non-DeformCaps*), and a version which removed the routing procedures (*No-Routing*). These ablations show the contribution of each component of the proposed method.

| Method | Backbone | FPS | AP | AP50 | AP75 | APS | APM | APL |
|---|---|---|---|---|---|---|---|---|
| MaskRCNN (He et al., 2017) | ResNeXt-101 | 11 | 39.8 | 62.3 | 43.4 | 22.1 | 43.2 | 51.2 |
| Deform-v2 (Zhu et al., 2019) | ResNet-101 | - | 46.0 | 67.9 | 50.8 | 27.8 | 49.1 | 59.5 |
| SNIPER (Singh et al., 2018) | DPN-98 | 2.5 | 46.1 | 67.0 | 51.6 | 29.6 | 48.9 | 58.1 |
| PANet (Liu et al., 2018) | ResNeXt-101 | - | 47.4 | 67.2 | 51.8 | 30.1 | 51.7 | 60.0 |
| TridentNet (Li et al., 2019) | ResNet-DCN | 0.7 | 48.4 | 69.7 | 53.5 | 31.8 | 51.3 | 60.3 |
| YOLOv3 (Redmon et al., 2018) | DarkNet-53 | 20 | 33.0 | 57.9 | 34.4 | 18.3 | 25.4 | 41.9 |
| RetinaNet (Lin et al., 2017) | ResNeXt-FPN | 5.4 | 40.8 | 61.1 | 44.1 | 24.1 | 44.2 | 51.2 |
| RefineDet (Zhang et al., 2018) | ResNet-101 | - | 36.4 / 41.8 | 57.5 / 62.9 | 39.5 / 45.7 | 16.6 / 25.6 | 39.9 / 45.1 | 51.4 / 54.1 |
| CenterNet (Zhou et al., 2019a) | DLA-34 | 28 | 39.2 / 41.6 | 57.1 / 60.3 | 42.8 / 45.1 | 19.9 / 21.5 | 43.0 / 43.9 | 51.4 / 56.0 |
| **DeformCaps (Ours)** | DLA-34 | 15 | 38.0 / 40.6 | 55.5 / 58.6 | 41.3 / 43.9 | 20.0 / 23.0 | 42.5 / 42.9 | 51.8 / 56.4 |
| Non-DeformCaps | DLA-34 | 20 | 27.2 / 30.1 | 41.8 / 45.9 | 29.6 / 32.6 | 13.3 / 15.6 | 30.0 / 30.8 | 35.0 / 40.6 |
| No-Routing | DLA-34 | 15 | 35.9 / 38.9 | 52.7 / 56.0 | 39.1 / 42.4 | 17.8 / 20.9 | 40.6 / 42.0 | 48.2 / 54.4 |

Table 1: Single-scale / multi-scale results on COCO test-dev. Top: Two-stage detectors; Middle: One-stage detectors; Bottom: Ablation studies on deformable capsules and SE-Routing.

---

[1]We will make our code publicly available for the community for reproducible research.

## 5 RELATED WORKS

### 5.1 CAPSULE NETWORKS

The idea of capsules was first introduced by Hinton et al. (2011). Sabour et al. (2017) extended this and proposed dynamic routing between capsules. The EM routing algorithm was then modified by Hinton et al. (2018). Recently, capsule networks could achieve the state-of-the-art performance for a wide range of applications: video object segmentation (Duarte et al., 2019), point cloud segmentation (Zhao et al., 2019), explainable medical diagnosis (LaLonde et al., 2020), text classification (Zhao et al., 2018), sentiment analysis (Wang et al., 2018), and various other applications (Vijayakumar, 2019).

### 5.2 OBJECT DETECTION

**Region proposal-based approaches:** R-CNN was one of the first successful deep object detectors, in which a selective search algorithm was used to select a number of region proposals, CNN features were extracted from each of the region proposals and were used to both classify the object and regress its bounding box (Girshick et al., 2014). The later addition of Fast R-CNN (Girshick, 2015) provided end-to-end training and addressed the speed and efficiency issues of R-CNN.

**Anchors-based approaches:** Anchors-based approaches sample fixed-shape bounding boxes (anchors) around a low-resolution image grid, then attempt to classify anchors into object classes. Faster R-CNN (Ren et al., 2015) generates region proposals in a first stage network, then attempts to classify and regress bounding boxes for the top-$k$ highest scoring anchors in a second stage network. Later studies such as Redmon et al. (2016) dramatically speed up the process by converting the proposal classifier to a multi-class one-stage detector. Since then, researchers have been working on improving one-stage detectors by including shape priors (Redmon & Farhadi, 2017; 2018), multiple feature resolutions (Liu et al., 2016), re-weighting the loss among different samples (Lin et al., 2017), or modeling channel-wise attention (Chen et al., 2020).

**Keypoint estimation-based approaches:** CornerNet (Law & Deng, 2018) attempts to detect objects by predicting two bounding box corners as keypoints. ExtremeNet (Zhou et al., 2019b) extends CornerNet's approach by estimating all corners and the center of the objects' bounding box. However, these methods rely on significantly slow combinatorial grouping post-processing stage. Zhou et al. (2019a) proposed CenterNet which attempts to predict only an objects' center point, and regress all other necessary values from there without the need for grouping or post-processing.

## 6 DISCUSSIONS, LIMITATIONS, & FUTURE WORK

Our proposed deformable capsules (*DeformCaps*) with *SplitCaps* object-class representations and Squeeze-and-Excitation inspired *SE-Routing* algorithm represents an important step for capsule networks to scale-up to large-scale computer vision problems, such as object detection or large-scale classification. Our proposed one-stage object detection capsule network is able to obtain results on MS COCO which are on-par with other state-of-the-art one-stage CNN-based networks for the first time in the literature, while also producing fewer false positives. Examining the qualitative results, provided in Appendix A.3, lends empirical evidence that *DeformCaps* can better generalize to unusual poses/viewpoints of objects than CenterNet (Zhou et al., 2019a). We hope our work will inspire future research into the considerable potential of capsule networks.

**Limitations:** Our study contains some limitations, discussed in greater detail in Appendix A.1. Briefly, (1) we had difficulty integrating the bounding box regression values into our capsule object detection head; (2) the choice of descriptors used in the squeeze is somewhat handcrafted, and is open to further investigation; (3) the choice of dimensions to model the class-agnostic instantiation parameters of objects was chosen semi-arbitrarily and could likely improve from fine-search; and (4) the choice of reconstructing objects' masks versus image patches is not thoroughly explored.

**Future directions:** The reconstruction sub-network of *DeformCaps* could possibly be trained to produce a fast single-shot instance segmentation framework. At test, potentially detected objects could have their instantiation vectors reconstructed into objects' masks, then these masks would simply be resized to the predicted bounding-boxes, similar to He et al. (2017) but without needing to have the initial reshape and ROI alignment required in their two-stage approach.

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

# A  APPENDIX

## A.1  EXTENDED EXPLANATIONS OF LIMITATIONS AND POSSIBLE RECOMMENDATIONS

In the discussion section of the main body of our paper, we mentioned four potential limitations in our study. We would like to discuss these in a bit more detail here. Since so many components of our method are newly introduced, there is a wide range of choices which could be investigated and improved by future researchers and engineers, and we suggest a few of those here:

**(1)** We had difficulty in integrating the bounding box regression values into our capsule object detection head. In our implementation, the class-agnostic capsules are trained to predict scale-normalized masks of $28 \times 28$. Ultimately, we would like to integrate predicting the object masks and the boxes for those masks together, as these tasks surely share mutual information. However, to the best of our knowledge, no published works exist for using capsules on a real-valued regression task.

**(2)** For our proposed SE-Routing, as with the original Squeeze-and-Excitation network, the choice of descriptors computed in the squeeze is somewhat handcrafted. We propose to use the cosine angle, KL divergence, and variance, and provide justifications for each of these choices, then allow the excitation to learn which of these pieces of information is most beneficial dynamically for each given input. Nonetheless, it is completely plausible that different descriptors could yield superior results. We unfortunately do not have the compute resources to run ablation studies over each of these chosen descriptors individually.

**(3)** The choice of 64 dimensions to model the class-agnostic instantiation parameters was decided somewhat empirically. As we argued in the main paper, it is unlikely that all variations across object poses are completely class independent; thus, to represent these extra dimensions of variation, we increase our vector lengths considerably ($16 \to 64$). However, it is possible that the number of class-independent and class-dependent variations is significantly higher or lower than the value chosen, and largely will depend on the complexity of the data being modeled. This difficulty is analogous to determining the optimal number of convolutional filters to use at every given layer of a CNN. Related to this, there is the potential for the class-dependent dimensions of the instantiation vectors to have unwanted influence over the cosine angle descriptors when attempting to represent objects of other classes. It could be beneficial to pass class information from the class presence capsule type over to the object instantiation capsule type to dynamically attend to the relevant dimensions of its vector for a given object. In a similar manner, it could be beneficial when computing the probability aggregation using the linear opinion pool to weight the expert opinions in proportion to their uncertainty instead of uniformly.

**(4)** We chose to reconstruct object's masks with the motivation of forcing the network to learn variations in shape, pose, and deformations. Since CNNs are known to be biased to texture information

over shape, we chose not to explicitly supervise the learning of any texture information. Nonetheless, it is plausible that reconstructing the object with texture could yield superior performance. Further, we chose to set the value of the reconstruction regularization's contribution to the loss to 0.1, following what was found most beneficial by CenterNet (Zhou et al., 2019a) for weighting the size loss contribution, and from a concern to not over-regularize the network early in training, then stepped this value to 2.0 half-way through training to make its value roughly equal to the other loss terms. From our experience, the accuracy remained fairly consistent across values up to 2.0 for this term, while setting its weight to 0.0 resulted in a degradation of performance. We found that increasing the value during training led to faster improvements in performance, consistent with other works in the literature that use such a regularization term. Engineering efforts on this parameter, such as a temperature function to automatically increase this weight during training, may prove beneficial if the goal is to reach the maximum possible accuracy.

## A.2 ANALYSIS OF FALSE POSITIVES

DeformCaps tends to be more conservative with its detections than CenterNet. This can be observed both by the slightly lower confidence scores (typically 0.1 less than CenterNet for most detections), and by the overall fewer amount of boxes placed in scenes. CenterNet tends to produce far more false positives than DeformCaps, both in the case of incorrect detections and of multiple detections for the same object which failed to be suppressed by the NMS algorithm. Though DeformCaps producing slightly lower confidence scores might account for some of the reduction in false positives, we observe CenterNet consistently producing fairly confident false predictions while DeformCaps does not produce a detection in the same region at all (see qualitative examples in Appendix A.3). A quantitative analysis of this is provided in Table 2. These number are generted using the official MS COCO evaluation code in it's standard operation. However, instead of only returning the average precision (AP) ratio of true positives (TP) and false positives (FP), namely $TP/(TP + FP)$, we also return the raw FP count as well.

|  | AFP 50:95 | AFP 50 | AFP 75 |
|---|---|---|---|
| **CenterNet** | 144.270 | 218.842 | 154.387 |
| **DeformCaps** | 139.516 | 209.465 | 149.904 |

Table 2: Comparison of mean average false positives (AFP) between the proposed DeformCaps and CenterNet at the standard IOU thresholds for MS COCO.

In the default operation of CenterNet, there is no non-maximum suppression (NMS) operation performed. Instead, a sort of pseudo-NMS is performed by passing a $3 \times 3$ Max Pooling operation over the detection grid activations to extract objects' centers. When running CenterNet with multi-scale testing, a NMS operation is then added to effectively *choose* which scale is the best fit for a given object detection. Therefore, the false positives being seen in Appendix A.3 are a direct results of multiple object centers being predicted incorrectly for the same object or object centers being predicted where there are no objects. We find that DeformCaps, which predicts objects as capsule vectors and not scalar, does not suffer from these former class of FPs.

This observation of less false positive detections is consistent with what we would expect from a capsule network with dynamic routing as compared to a traditional convolutional neural network (CNN). Where a CNN passes on all activations to the next layer, capsule networks utilize a dynamic routing algorithm to only pass on activations if there is agreement amongst the child capsule projections. In our proposed method specifically, with a SplitCaps structure and SE-Routing, the agreement is computed for projections of both the pose and class of the object being represented. It follows naturally that this would limit the amount of false positive detections which are produced, by reducing the amount of activations that get passed on. Further, we find from a survey of these qualitative examples that DeformCaps is better able to detect objects when being presented in an unusual pose or from an usual viewpoint than its CenterNet counterpart. This gives empirical support to one of the purported benefits of capsule networks, to be able to better generalize to unseen poses and viewpoints.

### A.3 Qualitative Results on MS COCO

Included in this section are a number of qualitative examples for CenterNet (Zhou et al., 2019a) and the proposed DeformCaps on the MS COCO test-dev dataset (Lin et al., 2014) using both flip and multi-scale augmentations. Results for CenterNet were obtained using the official code and trained models provided by the authors (Zhou et al., 2019a). While we do not make any sweeping claims, we wish to comment on a few general patterns that seemed to emerge in these examples. In Figure 3, we show a prototypical example of the general trend we are describing. In Figures 4– 5, we include more examples of this trend. In Figures 6– 7, we include a set of interesting examples of unusual object viewpoints or poses being better captured by DeformCaps than by CenterNet.

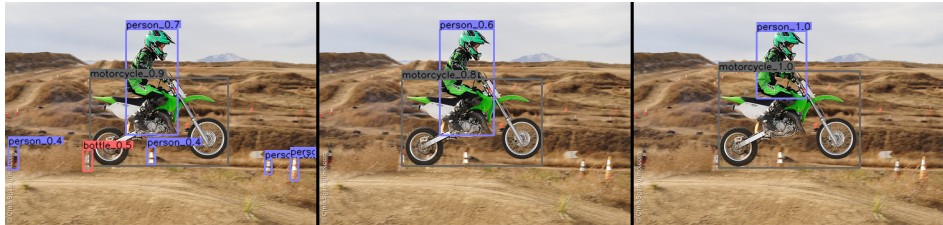

Figure 3: Qualitative example for CenterNet (Zhou et al., 2019a) (**leftmost**), the proposed Deform-Caps (**center**), and the ground-truth annotations (**rightmost**) on the MS COCO test-dev dataset. While CenterNet produces slightly higher average precision (AP) values than DeformCaps, it also seems to produce more false positives on average than DeformCaps.

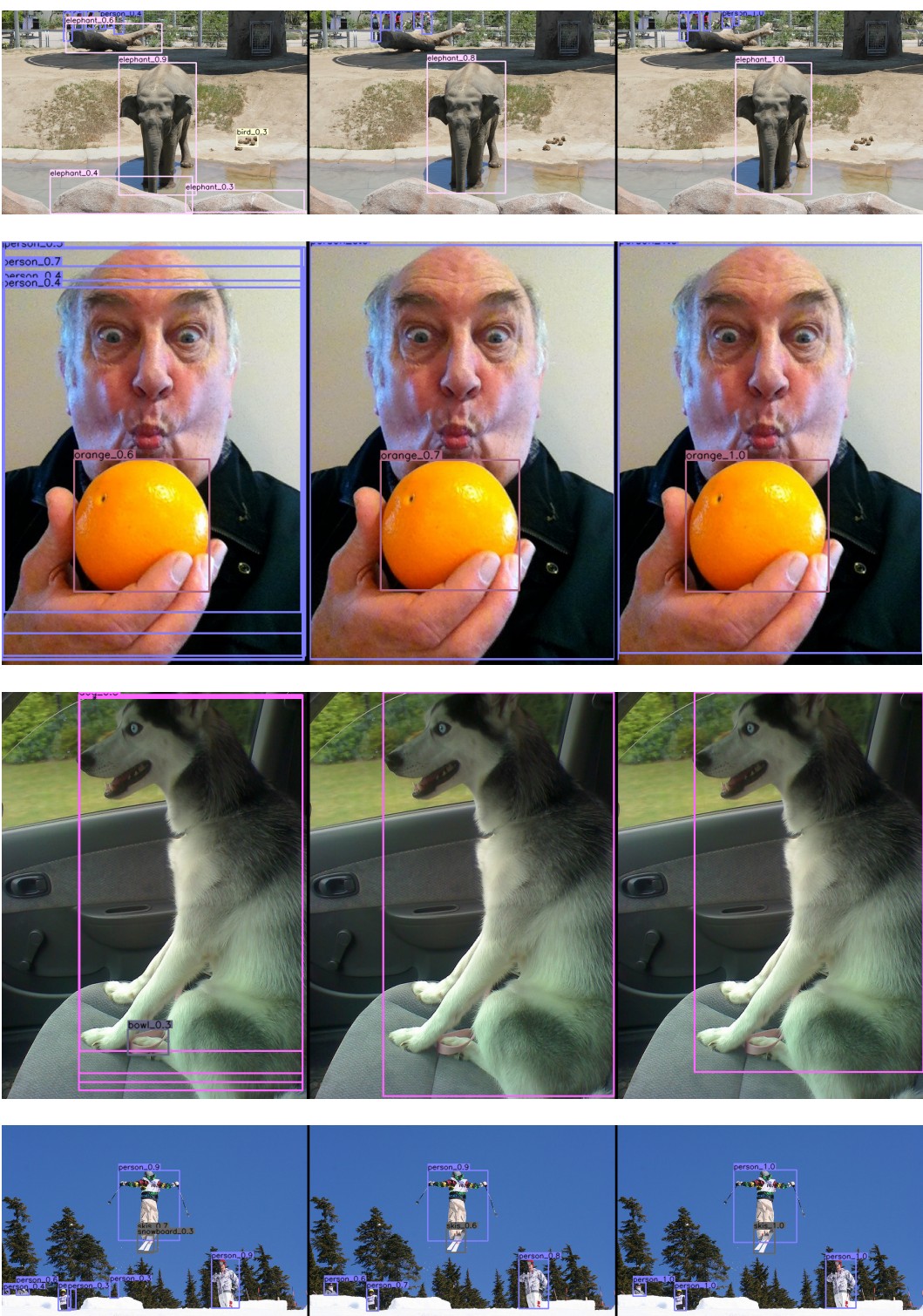

Figure 4: Qualitative examples showing CenterNet (Zhou et al., 2019a) (**leftmost**) consistently producing a higher number of *false positive detections* than the proposed DeformCaps (**center**), as compared to the ground-truth annotations (**rightmost**) on the MS COCO test-dev dataset.

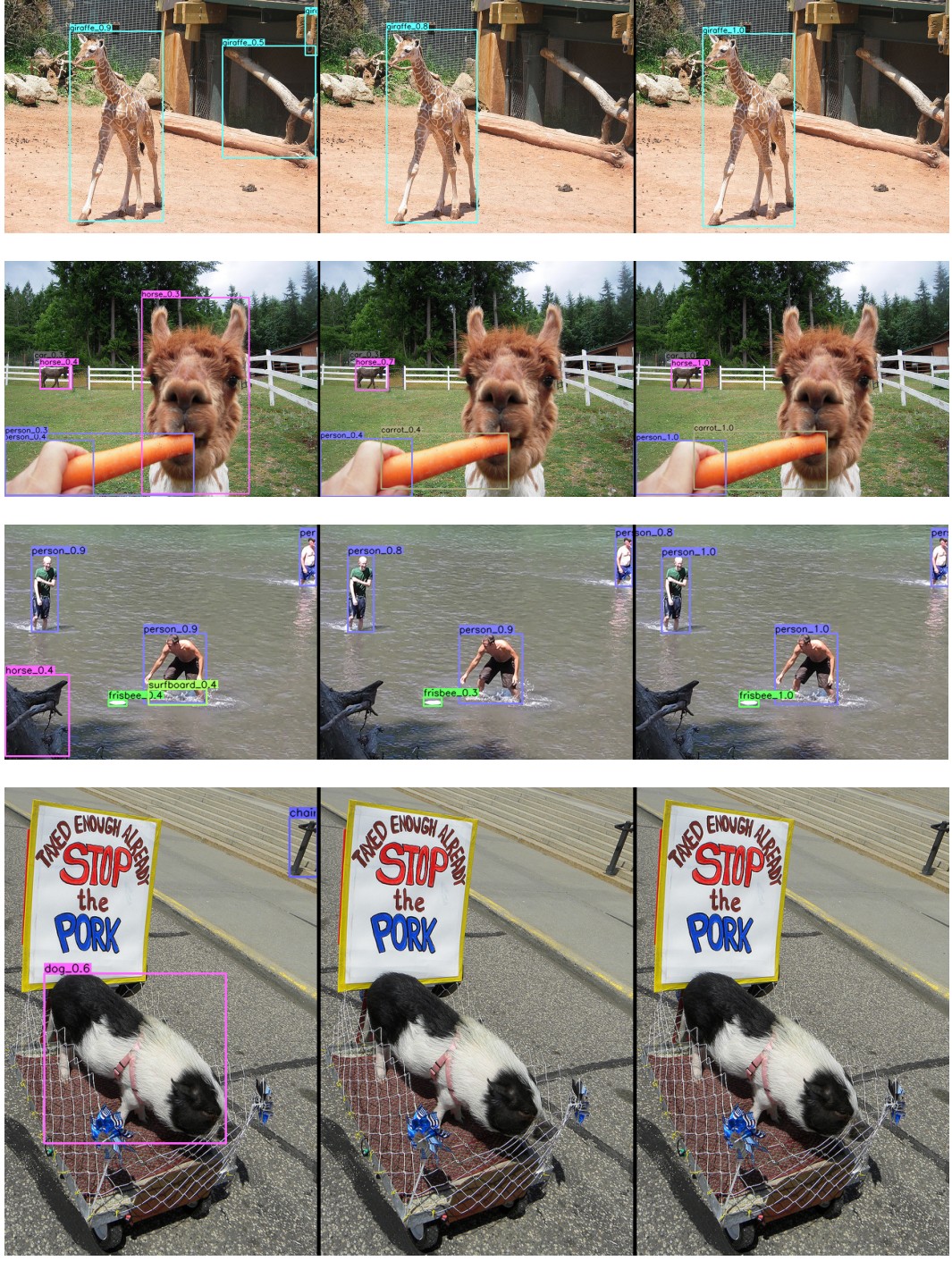

Figure 5: Some more qualitative examples showing CenterNet (Zhou et al., 2019a) (**leftmost**) consistently producing a higher number of *false positive detections* than the proposed DeformCaps (**center**), as compared to the ground-truth annotations (**rightmost**) on the MS COCO test-dev dataset.

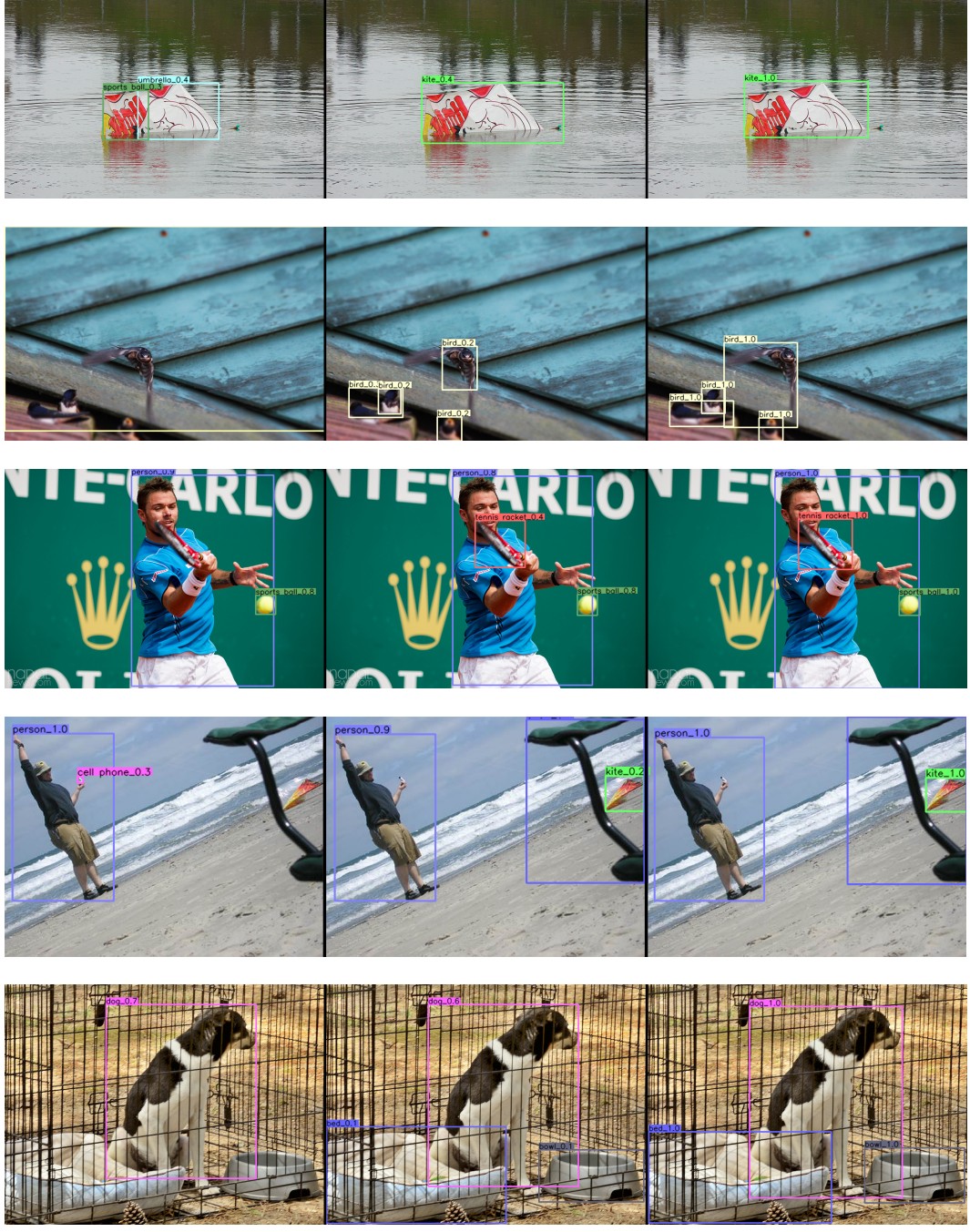

Figure 6: Qualitative examples showing CenterNet (Zhou et al., 2019a) (**leftmost**) struggling to detect objects being presented *in unusual poses or from unusual viewpoints*, while the proposed DeformCaps (**center**) successfully captures these cases, as compared to the ground-truth annotations (**rightmost**) on the MS COCO test-dev dataset.

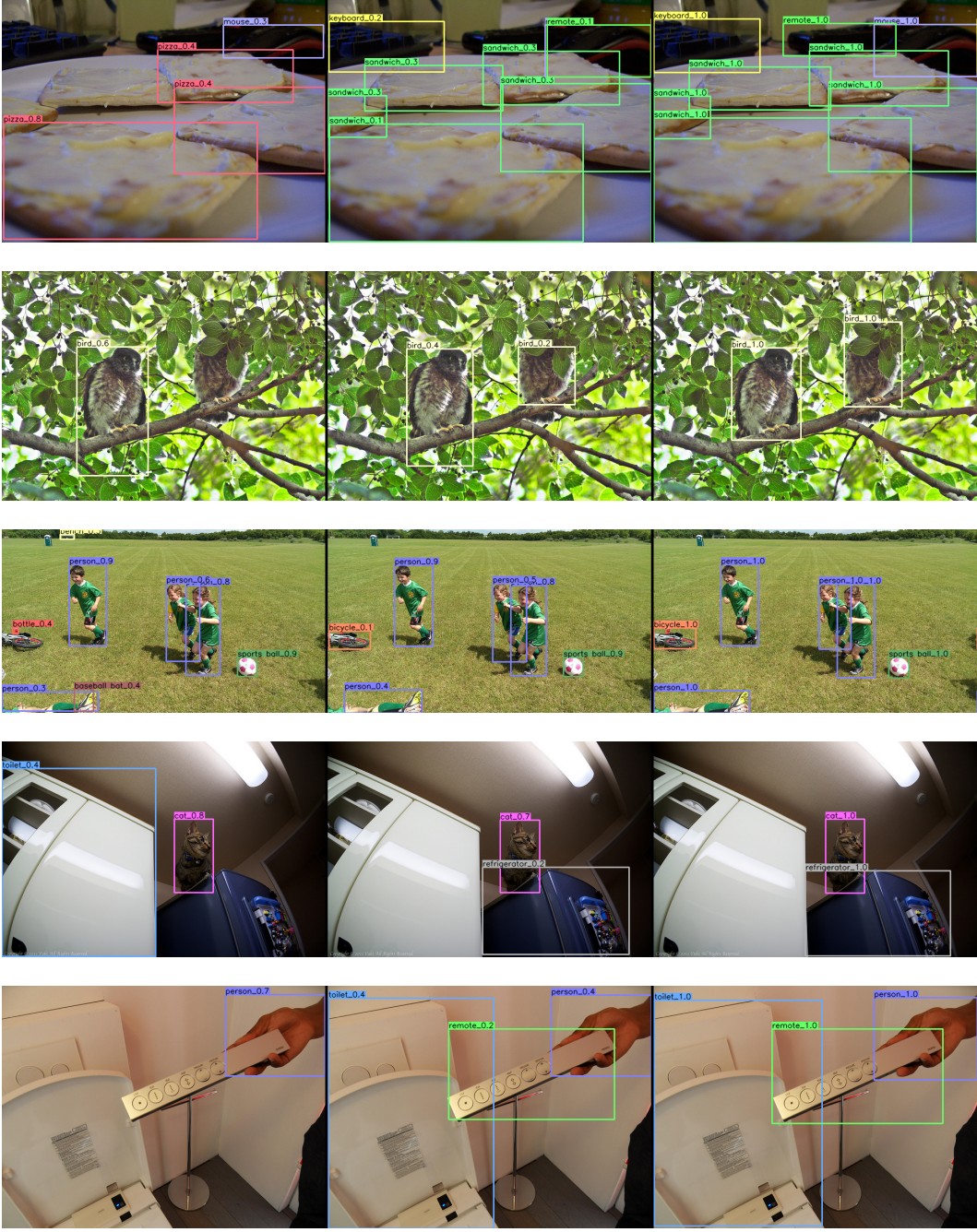

Figure 7: A few more qualitative examples showing CenterNet (Zhou et al., 2019a) (**leftmost**) struggling to detect objects being presented in *in unusual poses or from unusual viewpoints*, while the proposed DeformCaps (**center**) successfully captures these cases, as compared to the ground-truth annotations (**rightmost**) on the MS COCO test-dev dataset.

