# OpenReview forum: "Deformable Capsules for Object Detection"
_ICLR.cc/2021/Conference — Reject_

### Official Review · AnonReviewer2 · 2020-10-27
**A good subminssion that can be improved further**

**Rating:** 6
**Confidence:** 4

**Review:**

Pros:
It is indeed that capsules have many promising attributes but they are not quite easy to be exploited in object detectors. This paper has addressed many problems existed when applying capsule architectures for the object detection task. In particular, deformable capsules, SplitCaps, and SE-Routing are respectively introduced to help tackle the object detection with capsules. I believe the techniques of this paper will attract interests from researchers in the corresponding areas. The writing is also good.

Cons:
There are several points that prevent me from giving a higher rating:
1) I feel that the descriptions or presentations of introduced techniques are not quite sufficient. In particular, I am confused about the details of deformable capsules. What are the exact operations performed by the deformable capsules? Is it to make parent capsules only aggregate information from a smaller set of their children and the sampling of such a smaller set is implemented by deformable operations? I believe it will be much better to present some detailed formulations or equations to illustrate this operation. Similarly, I also recommend adding detailed equations to describe SE-Routing.

2) Without sufficient information about operations, I found that the descriptions of motivations are also quite weak, especially in the introductory part. It seems that the authors only mention the problems they will address and the techniques they proposed to address problems. There is not sufficient content briefly explaining why the proposed techniques can tackle the mentioned problems, or at least what are the advantages of the proposed techniques for tackling the problems.

3) The figures also have many confusing points. For example, in Figure 1, the authors marked that solid red arrows represent 'up' but I can only find dotted red arrows in the figure. Moreover, in what place the modules inside the big blue box is implemented in the detector? After reading, I think it should represent what's inside the SplitCaps, but I do recommend the authors to add some indicative symbols to corresponding related modules.

4) In addition to the reviewed literature, I found that there are some missing articles that also study how to borrow the capsule concepts for various computer vision tasks. For example:
[1] Vijayakumar T. Comparative study of capsule neural network in various applications[J]. Journal of Artificial Intelligence, 2019, 1(01): 19-27.
[2] Zhao Y, Birdal T, Deng H, et al. 3D point capsule networks[C]//Proceedings of the IEEE conference on computer vision and pattern recognition. 2019: 1009-1018.
[3] Chen Z, Zhang J, Tao D. Recursive Context Routing for Object Detection[J]. International Journal of Computer Vision, 2020: 1-19.

Overall, I would like to have some feedback from the authors regarding the above issues before making my final decision.

---

> ### Author Response · Authors · 2020-11-22
> **Response to Reviewer 2 [part 1]**
>
> We appreciate the reviewer’s comments. We will do our best to address each of these comments below in a succinct manner.
>
> 1. “I feel that the descriptions or presentations of introduced techniques are not quite sufficient. In particular, I am confused about the details of deformable capsules. What are the exact operations performed by the deformable capsules? Is it to make parent capsules only aggregate information from a smaller set of their children and the sampling of such a smaller set is implemented by deformable operations? I believe it will be much better to present some detailed formulations or equations to illustrate this operation. Similarly, I also recommend adding detailed equations to describe SE-Routing.”
>
> We understand the reviewer’s request for clarification on some of the points related to deformable capsules specifically and the work in general. As mentioned in our response to Reviewer 1, capsule networks by nature can be somewhat involved in their explanation. We did our best now to sample the relevant capsule literature in the field at major venues (e.g. NeurIPS, ICLR, etc.) and ensure that our basic definitions and formulations are consistent with those already accepted and published in the field.
>
> To directly address the raised questions, the purpose of deformable capsules in short is this. The original method of fully-connected capsules in CapsNet [a] was to sample all children from a previous layer when forming parents in the current layer. This method does not scale up to larger image sizes (described in Section 2 Paragraphs 1 and 4), so several works introduced convolutional style capsules which only sample children from a smaller local kernel centered on the parent capsules spatial location (described in Section 2 Paragraph 1). Our approach is an extension of these convolutional methods which subsample children. However, instead of sampling in a geometrically-rigid manner, which does not easily allow for spatial deformations of objects and is antithetical to one of the main benefits of capsules (described in Section 2 Paragraph 1), we do so in a learned dynamic manner (described in Section 2 Paragraph 2). The formulation of that is identical to the formulation in deformable convolutions, where a first set of weights is learned to dynamically predict a set of offsets for the second set of weights to be applied to (cited and described in Section 2 Paragraph 3). We choose not to repeat the entire formulation of deformable kernels in the paper, as such is fairly well-known already. In regards to equations for SE-Routing, we have included a large number of in-line equations throughout Section 3.2 to precisely provide the formulation of our proposed methods.

---

> > ### Author Response · Authors · 2020-11-22
> > **Response to Reviewer 2 [part 2]**
> >
> > 2. “Without sufficient information about operations, I found that the descriptions of motivations are also quite weak, especially in the introductory part. It seems that the authors only mention the problems they will address and the techniques they proposed to address problems. There is not sufficient content briefly explaining why the proposed techniques can tackle the mentioned problems, or at least what are the advantages of the proposed techniques for tackling the problems.”
> >
> > We are sorry there was some confusion about the motivation for our methods and how those methods solve the problems named. If the reviewer is asking about what are the benefits of capsule networks in general, and how exactly do capsule networks better represent objects and better generalize to unseen poses and viewpoints, we feel this is unfortunately outside of the scope of this work and point the reviewer to works such as [a-c]. If the reviewer is asking about the motivation for each component proposed in this paper and how they solve the referenced problems, we describe (and update them now in the revised manuscript) these in the body of the paper, within each corresponding section. We will summarize here too:
> >
> > Deformable capsules solves one of two problems in previous methods:
> > 1) Enables scaling of capsule network to high dimensional images, not possible with fully connected capsules; and
> > 2) Removes the geometric-constraints imposed by the previously used convolutional capsules aimed at solving this first issue.
> > The experiments on MS COCO show removing this geometric constraint was critical to achieving good performance on the object detection task where modeling the poses and deformations of objects is important in such a varied task and dataset.
> >
> > Next, SplitCaps allows capsules to scale-up to large numbers of classes and object detection for the first time ever in the literature by learning class agnostic object instantiation parameter vectors and class dependent activations. Previous capsule methods represent each class with their own capsules, preventing capsule networks from scaling up to larger tasks.
> >
> > Lastly, the new SplitCaps head structure precludes the ability to apply any existing dynamic routing methods. Further, previous dynamic routing methods were iterative, slow, and often non-convergent. Our proposed SE-Routing method is non-iterative and provides a way for dynamic routing to be performed in our new SplitCaps head in a manner which accounts for both class agnostic parameters and class-wise activations. The specific choices made within SE-Routing for the squeezes are all justified with explanations and citations and the results of the ablation study on this component shows it’s benefit to performance.
> >
> > 3. “The figures also have many confusing points... solid red arrows represent 'up' but I can only find dotted red arrows... recommend the authors to add some indicative symbols to corresponding related modules”
> >
> > We thank the reviewer for identifying these points of confusion, and to address this comment, we have updated the figure to fix these issues. The red arrow was made dashed in the key of and we updated the title of the blue box to show where the blue box fits into the overall architecture.
> >
> >
> > 4. “In addition to the reviewed literature, I found that there are some missing articles that also study how to borrow the capsule concepts for various computer vision tasks. For example:...”
> >
> > We unfortunately could not include all of the capsule papers we would like due to space limitations, but we agree that these are great candidates for inclusion. With the extra page allowed at this time, we have added these references. We appreciate the reviewer drawing these articles to our attention.
> >
> > Refs:
> > [a] Sabour, Sara, Nicholas Frosst, and Geoffrey E. Hinton. "Dynamic routing between capsules." Advances in neural information processing systems. 2017.
> > [b] Hinton, Geoffrey E., Sara Sabour, and Nicholas Frosst. "Matrix capsules with EM routing." International conference on learning representations. 2018.
> > [c] Punjabi, Arjun, Jonas Schmid, and Aggelos K. Katsaggelos. "Examining the Benefits of Capsule Neural Networks." arXiv preprint arXiv:2001.10964 (2020).

---

### Official Review · AnonReviewer3 · 2020-10-29
**Impressive results, would be nice to have more ablations**

**Rating:** 6
**Confidence:** 3

**Review:**

The paper proposes to use the capsules to perform object detection on COCO. Capsules, while showing promises, are usually too expensive for tasks beyond MNIST and Cifar. The authors propose three key improvements in DeformCaps, SplitCaps and SE-Routing to improve the efficiency and therefore allow capsules to be applied on larger tasks such as object detection. The authors claim novelties in:

-- First ever capsule-based object detection on COCO.

-- Deformable capsules that let parent to samples child capsules instead of having fixed connections to improve efficiency.

-- SplitCaps: a reparametrization trick to reduce the number of capsules needed.

-- Squeeze-and-excitation routing that finds agreement without iterative loop.

I agree with the authors' novelty claims.

Strengths:

-- Impressive results: Capsule Networks are well known to be computationally expensive. Getting it to work on large images is certainly no easy task. There has been other works operating on larger images, as noted by this paper, but they usually cannot achieve quality on par with CNNs. Having said that, I need to note that the SOTA for COCO has consistently improved, and is at ~51% AP [a], quite better than the numbers listed in Table 1.

-- Good novelty: DeformCapsule and SplitCaps both follow a straightforward pattern to dissect prohibitively large tensor predictions into smaller ones. And together with the SE-routing methods, the novelty should be sufficient for this venue.

-- Clear writing. Figure 1 and 2 helped a lot with the understanding of the key components as well.

[a]: SpineNet: Learning Scale-Permuted Backbone for Recognition and Localization

Weaknesses:

-- Insufficient experimentation: There is only a single experimental table that contains both comparisons to the SOTA and ablations to DeformCaps and SE-Routing. It would be nice to see additional results, be it on another dataset / task (e.g. sem seg), or even additional ablations. For example, one premise of capsules was the requirement of less training data. Is it true here as well?

Conclusion:

I'd recommend acceptance because of the strong results and good novelty. The only major weakness is the lack of additional ablations to highlight the advantages (if not on AP) over CNNs. Nonetheless, I believe this work is a good step towards more interests and advances in capsules, and will be of interests to the audience of this venue.

Post-rebuttal:
I concur with R1 in that the results look poor when taking into the account how close the proposed method is compared to CenterNet and that it was finetuned from a CenterNet. Therefore, I'll lower my rating to 6.

---

> ### Author Response · Authors · 2020-11-22
> **Response to Reviewer 3**
>
> We appreciate the reviewer’s thoughtful consideration of our work and agree with the reviewer’s conclusions. We are hopeful the novelties proposed, which culminate in enabling capsules to scale up to large-scale tasks, will encourage interest and future advancements in capsule networks for a wide range of applications. For completeness, we would like to address two of the points made by the reviewer.
>
> 1. “SOTA for COCO has consistently improved, and is at ~51% AP [a]”.
>
> The field is moving extremely quickly and results are consistently being improved. A main advantage of our proposed capsule innovations is that they act solely within a new detection head on the network. Hence, any improvements in network backbones can be easily swapped in. For example, the reviewer referenced SpineNet [a] uses a RetinaNet detection head which could easily be replaced by our proposed SplitCaps detection head. This is a strength of our work, indeed.
>
> 2. “It would be nice to see additional results, be it on another dataset / task (e.g. sem seg), or even additional ablations. For example, one premise of capsules was the requirement of less training data. Is it true here as well?”
>
> We again completely agree with the reviewer’s opinion. Additional experimentation is nearly always desirable and we can think of a large number of new datasets and tasks we are excited to pursue, as well as hopefully see others in the community investigate. For example, the task of ReID/Search from different viewpoints would be an excellent candidate for testing the generalization of capsules to new viewpoints. However, we would like to emphasize that the large number of interesting further studies that could be performed is a benefit of our proposed work and will hopefully lead to a flourishing of new capsule network research, rather than a negative for us not performing all of them ourselves in this single-scoped paper which introduces these concepts. Other works, such as [b] demonstrate capsule networks require less training data, and we felt such ablations about capsule networks in general would take away from the specific innovations being proposed in this study.
>
> Refs:
> [a] Du, Xianzhi, et al. "SpineNet: Learning scale-permuted backbone for recognition and localization." Proceedings of the IEEE/CVF Conference on Computer Vision and Pattern Recognition. 2020.
> [b] Jiménez-Sánchez, Amelia, Shadi Albarqouni, and Diana Mateus. "Capsule networks against medical imaging data challenges." Intravascular Imaging and Computer Assisted Stenting and Large-Scale Annotation of Biomedical Data and Expert Label Synthesis. Springer, Cham, 2018. 150-160.

---

### Official Review · AnonReviewer1 · 2020-10-30
**An interesting idea of using capsule network for object detection; The experimental sections are relatively weak and cannot support the claim.**

**Rating:** 4
**Confidence:** 3

**Review:**

### summary
This paper introduces capsule network for object detection. To solve the issue of capsule network when applied to large-scale detection problems, this paper develops deformable capsules, a new  prediction head SplitCaps, and a dynamic routing algorithm, SE-Routing. Experiments are conducted on COCO where it performs slightly worse than the baselines but arguably predicts less false positives.

### pros
I appreciate the spirit of developing new architectures for a highly-competitive task like object detection. It's totally acceptable that the performance cannot fully match the SOTA results in this kind of scenario. This paper is insightly as it studies several important issues of capsule network when being applied to large-scale visual tasks. The proposed solutions have at least make the framework run-able on standard hardware.

### cons
1. My current rating is not purely based on the performance, but it is one of the major concerns here. The proposed framework (1) performs worse than baseline -- CenterNet, and (2) performs much worse then the SOTA solutions (Tab.1) in terms of both performance and speed. Note that the network weights are initialized from CenterNet.
2. The whole analysis in Appendix A.2 reads very confusing for several reasons:
 * CenterNet achieves higher AP (TP/(TP+FP)) and also high FP, which is a quite common thing to me. Ignoring the ratio and simply compare the absolute value of FP doesn't seems to be the correct thing to do.
 * Some visualizations are weird like Fig.4 human and dog. Why NMS cannot remove such highly-redundant predictions? It seems more like a bug to me.
 * On page 12 top row it writes "CenterNet consistently producing fairly confident false predictions (e.g. > 0.4)", how does this threshold 0.4 is chosen? It's not a standard thing and in fact if we improve it to 0.5 or higher, most of the FPs about CenterNet will disappear. Moreover, is this how FP gets counted? If yes, I think this is problematic as the total number (TP+FP) is changing across methods. A more fair way is to choose top-k (k=100 in coco) predictions and compute the metrics.
3. Lots of reference are missing. For example, Tab.1 compares multiple different methods but most of them are not cited. The related work only cites few very classical papers, but most recent works are missing.
4. The writing, especially the approach section, isn't very clear. I had a relatively hard time to understand how do each modules work, and what are the motivations behind these design choices.
5. Moreover, there are couple of new things proposed, and each one of them have specific design choices and parameters. However, there are no ablation studies properly studying them. As a results I don't understand which module is working, and how does it work.

### questions
1. How do the FPS numbers in Tab.1 get computed? Are they simply borrowed from original paper or some published work? Some of them look strange to me.
2. Why KL in Sec.3.2? Does the asymmetric natural of KL influence the results?

---

> ### Author Response · Authors · 2020-11-22
> **Response to Reviewer 1 [part 1]**
>
> We appreciate the reviewer’s thorough and critical review of our work. After careful inspection of each critique, we have identified that the major concern is related to a claim which is not made (performance) in our paper. Furthermore, several areas of potential misunderstanding for readers were highlighted by this review and we appreciate the chance to make these more clear.
>
> We will address each of these points individually; but first, we would like to re-emphasize the contribution of this study is the development of novel techniques and architectures which allow capsule networks to scale up to large scale tasks and object detection for the first time in the literature. Our goal was not to push state-of-the-art performances and we are not claiming any such thing in the paper. Similar to the CapsNet [a] paper (cited more than 2800 times since 2017), which formalized the concept of capsules and introduced dynamic routing, but produced less than state-of-the-art results on CIFAR-10. In analogy to CapsNet [a] paper, we feel that our study is an important study opening many potential avenues for further investigation of capsules on a now much wider range of accessible tasks.
>
> 1. “ The proposed framework (1) performs worse than baseline -- CenterNet, and (2) performs much worse then the SOTA solutions (Tab.1) in terms of both performance and speed.”
>
> We kindly direct the reviewer to our above paragraph about the primary focus of our study and also suggest that the reviewer’s point (2) is perhaps a bit of an overstatement. While we perform inferior to very slow networks such as TridentNet which runs at 0.7 FPS., we still outperform networks which run at comparable speeds such as Mask R-CNN which runs at a slower 11 FPS (compared to our 15 FPS) and achieves a lower mAP score of 39.8 compared with our 40.6. We also perform 7.6 mAP points better than YOLOv3 while only being 5 FPS slower (15 compared to 20 FPS). While we are obtaining all these encouraging results, we would like to re-emphasize that our focus was not accuracy improvement or architectural engineering to boost the accuracy.
>
> 2. Several comments related to False Positives appendix, “Appendix A.2 reads very confusing for several reasons”
>
> 2.a. “CenterNet achieves higher AP (TP/(TP+FP)) and also high FP, which is a quite common thing to me. Ignoring the ratio and simply comparing the absolute value of FP doesn't seem to be the correct thing to do.”
>
> We acknowledge that CenterNet produces slightly higher AP scores on average than our proposed Deformable Capsules, and we do not ignore this ratio, and in fact we stated it in the main Table 1. Many applications exist where reducing false positives is of extreme and primary importance (e.g. military or security applications). In these applications, users are willing to sacrifice some missed detections and even a slightly lower AP if it means significantly reduced FPs. This is why we add the additional reporting of not only the ratio TP/(TP+FP) but also the raw FP values as well.
>
> 2.b. “Some visualizations are weird like Fig.4 human and dog. Why NMS cannot remove such highly-redundant predictions? It seems more like a bug to me.”
>
> CenterNet and other anchorless detection frameworks are not that easy to interpret compared to other detection frameworks; hence, we appreciate this comment from the reviewer that we now have a chance to resolve this understandable confusion. We have updated the language in the paper to better explain where these false positives are coming from, and how our proposed method helps to solve this issue. In short, CenterNet performs a sort of pseudo-NMS by passing a 3x3 Max Pool over the final detection grid to extract objects’ centers, and only performs a true NMS when running multi-scale testing (note: we are using multi-scale testing in Fig.4). However, in CenterNet (and we emphasize we are using the official CenterNet code repo), NMS acts by choosing which scale is the most relevant for each detected objects’ centers and does not remove separately detected objects which might overlap significantly. This is a design choice since overlapping boxes can occur naturally, and CenterNet relies on the detection centers to determine the existence of objects. Thus, if multiple center points are incorrectly predicted for the same object, this will not be suppressed by the NMS operation. While it is true that further post-processing could likely be performed to help mitigate this issue, our proposed method removes the issue completely. Our SplitCaps prediction head produces a detection grid which is a set of capsule vectors rather than scalars, with the final predictions having passed through the proposed SE-Routing dynamic routing algorithm. This combination of techniques produces a final detection grid which, after passing through the 3x3 Max Pool, does not contain the same high degree of redundant detection centers for objects.

---

> > ### Author Response · Authors · 2020-11-22
> > **Response to Reviewer 1 [part 2]**
> >
> > 2.c. “On page 12 top row it writes "CenterNet consistently producing fairly confident false predictions (e.g. > 0.4)", how does this threshold 0.4 is chosen?... A more fair way is to choose top-k (k=100 in coco) predictions and compute the metrics.”
> >
> > We apologize for creating confusion with this "(e.g. > 0.4)". We did not set any such thresholds and simply used the COCO evaluation as normal with the top-k where k=100, just as the reviewer suggests. The only difference is that in addition to reporting the ratio (TP / (TP + FP)) we also reported at what FP values that AP was obtained. The reason for including the e.g. > 0.4 was merely to show that the false positives being produced are not low-confidence false positives and thus are actually dragging the entire AP score down. We agree this makes things confusing and implies some sort of threshold. We have removed this confusing language and added more explicit details to the section to explain exactly how we obtained and are reporting the FPs.
> >
> > 3. “Lots of references are missing. For example, Tab.1 compares multiple different methods but most of them are not cited.”
> >
> > We apologize for this oversight on our part. We have since added citations for each of the methods in the table that we are comparing with.
> >
> >
> > 4. “The writing, especially the approach section, isn't very clear. I had a relatively hard time to understand how do each modules work, and what are the motivations behind these design choices.”
> >
> > We are sorry the reviewer had some difficulty here. Capsule networks by nature can be somewhat very involved in their explanation. After this comment, we did our best to sample the relevant capsule literature in the field at major venues (e.g. NeurIPS, ICLR, etc.) and ensure that our basic definitions and formulations are consistent with those already accepted and published in the field. Also, we would like to note that both Reviewer 2 and Reviewer 3 commented our writing was good and Reviewer 3 highlighted “DeformCapsule and SplitCaps both follow a straightforward pattern” and “Figure 1 and 2 helped a lot with the understanding of the key components”.
> >
> > 5. “Moreover, there are a couple of new things proposed, and each one of them have specific design choices and parameters. However, there are no ablation studies properly studying them. As a result, I don't understand which module is working, and how does it work.”
> >
> > To this point, we must ultimately disagree with the reviewer. The claim that “there are no ablation studies properly studying [the new things proposed]” is untrue. We are proposing three main components, namely DeformableCapsules, SplitCaps, and SE-Routing. Without SplitCaps, detection would not be computationally feasible and we provide the calculations showing this in Section 3.1 of the paper. For the other two components, we provide ablation studies in Table 1 which demonstrate their relative importance on the accepted scoring metrics. While it is true that further explorations of the design choices of each sub-component to these main contributions and all of their parameters could be investigated, and we agree such experiments would be beneficial, we do not feel it a reasonable request to require ablation studies for every design choice and every parameter. We highlight this limitation in Section 6 as well as Appendix A.1 of our study and give future investigators concrete directions for further study.
> >
> > Question 1: “How do the FPS numbers in Tab.1 get computed? Are they simply borrowed from original paper or some published work? Some of them look strange to me.”
> >
> > The FPS numbers are borrowed from the original CenterNet paper, where the authors took great efforts to produce results and FPS numbers all using the same hardware. We found that we were able to achieve nearly identical FPS values for CenterNet on our hardware, and thus felt safe borrowing the remaining values. This is noted in the paper where we stated, "Inference time on our hardware (IntelXeon E5-2687 CPU, Titan V GPU, Pytorch 1.2.0, CUDA 10.0, and CUDNN 7.6.5) was consistent with those reported by Zhou et al. (2019a).”
> >
> > Question 2: “Why KL in Sec.3.2? Does the asymmetric natural of KL influence the results?”
> >
> > KL Divergence is a commonly used method for comparing distributions. The reviewer is correct that the function is asymmetric; however, despite this, many neural networks find great success using KL Divergence, for example [b]. Most of the standard frameworks (e.g. PyTorch, TensorFlow, Keras) have both KL Divergence and KL Divergence Loss built in.
> >
> >
> > Refs:
> > [a] Sabour, Sara, Nicholas Frosst, and Geoffrey E. Hinton. "Dynamic routing between capsules." Advances in neural information processing systems. 2017.
> > [b] G. Seo, J. Yoo, J. Cho and N. Kwak, "Kl-Divergence-Based Region Proposal Network For Object Detection," 2020 IEEE International Conference on Image Processing (ICIP), Abu Dhabi, United Arab Emirates, 2020, pp. 2001-2005, doi: 10.1109/ICIP40778.2020.9191075.

---

### Decision · Program_Chairs · 2021-01-10
**Final Decision**

**Decision:**

Reject

**Comment:**

This work proposes capsule networks with deformable capsules for tackling object detection. All reviewers agreed that object detection is an important problem that is interesting to the ICLR community. Reviewers also agree that the proposed approach is novel and interesting, and in particular they mention that proposing an efficient capsule network for detection is non-trivial. On the less positive side, during the discussion phase all reviewers had concerns about the weak experimental validation, particularly missing ablation studies to analyse the effectiveness of their contributions. At the end, all reviewers felt that, while this is a promising direction of research for object detection, the experimentation should be improved.